# Prevention of Microsphere Blockage in Catheter Tubes Using Convex Air Bubbles

**DOI:** 10.3390/mi11121040

**Published:** 2020-11-27

**Authors:** Dong Hyeok Park, Yeun Jung Jung, Sandoz John Kinson Steve Jeo Kins, Young Deok Kim, Jeung Sang Go

**Affiliations:** School of Mechanical Engineering, Pusan National University, 2, Busandaehak road 63-2, Keumjeong-gu, Busan 609-735, Korea; dhpark90@pusan.ac.kr (D.H.P.); nextyj@pusan.ac.kr (Y.J.J.); stevejeokins@pusan.ac.kr (S.J.K.S.J.K.); jacksonydk@pusan.ac.kr (Y.D.K.)

**Keywords:** catheter, microspheres, blockage arching, convex air bubbles, slip, centrifugal

## Abstract

This paper presents a novel method to prevent blockages by embolic microspheres in catheter channels by using convex air bubbles attached to the channels’ inner wall surface. The clogging by microspheres can occur by the arching of the microspheres in the catheter. A few studies have been done on reducing the blockage, but their methods are not suitable for use with embolic catheters. In this study, straight catheter channels were fabricated. They had cavities to form convex air bubbles; additionally, a straight channel without the cavities was designed for comparison. Blockage was observed in the straight channel without the cavities, and the blockage arching angle was measured to be 70°, while no blockage occurred in the cavity channel with air bubbles, even at a geometrical arching angle of 85°. The convex air bubbles have an important role in preventing blockages by microspheres. The slip effect on the air bubble surface and the centrifugal effect make the microspheres drift away from the channel wall. It was observed that as the size of the cavity was increased, the drift distance became larger. Additionally, as more convex air bubbles were formed, the amount of early drift to the center increased. It will be advantageous to design a catheter with large cavities that have a small interval between them.

## 1. Introduction

Recently, a trend in the medical community is to achieve maximum therapeutic effect on diseases with minimal surgery and drug administration [1,2,3,4,5,6,7,8]. Medical science imaging has developed to find microscopic cancer cells and has enabled treatments such as a micro-invasive medical procedure without surgery. This procedure has minimal incisions and local anesthesia. Thus, it causes less pain and bleeding, resulting in a fast recovery.

Embolization is one of interventions carried out by the micro-invasive procedure using imaging equipment. A catheter is inserted to the correct vessel by digital subtraction angiography and organ positions. The artificial embolus microspheres are guided to block capillary vessels through which the myoma receives nutrients from the blood. This procedure is simpler and easier than other surgical methods and also has fewer treatment-related events. Figure 1 shows the schematic of fibroid embolization of microspheres with the catheter.

However, in the process of transporting embolic microspheres through a long and thin catheter with a diameter of 1 mm, the catheter often becomes blocked [9,10], and it is difficult to release the clogged microspheres in the catheter. In addition, if the microspheres are injected continuously despite the clogging, pressure builds up in the clogged part, and there may be a danger of bursting the capillary vessel due to the instant release of fluid and microspheres at high pressure. To this end, microsphere blockages in catheters must be resolved for proper embolism practices.

There may be two possible mechanisms on the blockage of microspheres flowing with a carrier liquid in the catheter [11]. One is the geometrical arching of the microspheres, and the other is the electrical deposition on the wall surface. The arching is a clogging phenomenon due to the steric effect in which the microspheres form an arch shape in the catheter channel [12,13,14]. Electrical deposition occurs when one microsphere attaches onto the surface of the channel wall due to electrical attractive force. Then, other microspheres are successively attached to the wall, and the attached microspheres form a lump gradually. The lump also gradually increases to block the catheter [15,16,17].

Many studies have been done on blockage, but few studies have introduced methods to solve the blockages. There are two representative methods to solve it. One is to remove the blockage of the microspheres with momentary motions by exerting oscillating waves [18,19,20,21]. This method constantly consumes power to use an amplifier and requires connecting the channel to a separate electrical device. The other method is to reduce the deposition of microspheres by inducing a repulsive electrical force chemically on the surface [22]. This requires an additional and sophisticated chemical treatment on the surface of the catheter.

This study proposes a novel method to prevent clogging by microspheres in a catheter by using convex air bubbles attached to the inner surface of the catheter. Convex air bubbles have a slip effect on their bubble surface and a centrifugal effect on the microspheres due to their small radius of curvature. In a fluid flow, the velocity is zero on the stationary wall, which is called the no-slip condition. However, the measurement of the micro-PIV(particle image velocimetry) over the surface of a convex air bubble shows that there exists a velocity with a certain value, which indicates the occurrence of the slip on it [23]. The slippery surface of the convex air bubble can reduce the friction, which has a key role in forming the blockage arching of microspheres in the catheter channel.

In addition, when the microsphere travels along the convex air bubble, centrifugal acceleration normal to the surface is obtained due to the microsphere’s small radius of curvature and is exerted on the microsphere. Even though it is a very low Reynolds flow, it makes the microsphere escape from the convex surface. Thus, the distance is increased between the microsphere and the channel wall. This centrifugal effect can reduce the attachment of microspheres so that it can prevent their deposition. As a result, both the friction reduction on the surface by the slip effect and the increased distance away from the catheter wall can effectively prevent the blockages caused by microsphere clogging.

To determine these effects by the convex air bubbles, straight catheter channels with microcavities were fabricated by the MEMS fabrication process. Additionally, a straight channel without cavities was fabricated as a reference, and the clogging of the microspheres was compared. The formation of the convex air bubbles in the cavities was visualized, and the two effects were experimentally examined and theoretically explained.

## 2. Fabrication of the Catheter Channel with Cavities

The term micro generally means a size from 1 μm to 1 mm. As the characteristic size becomes smaller, the ratio of the surface area to volume increases. Thus, the convex air bubbles attached in the microcavities hardly detach from the surface because the buoyancy and the shear force as a detaching force are much smaller than the surface tension as an attaching force [24].

The embolization catheter is generally cylindrical. The hydrophobic cylindrical tubes are manufactured by an extrusion process. However, in order to form the air bubble for the introduction of a liquid, the tube must have cavities on the inner surface. This makes the manufacturing process very complicated and challenging. In addition, it is very difficult to visualize the movement of the microspheres in the 3-dimensional cylindrical tube. So, for the experimental evaluation, the catheter channel with microcavities was fabricated precisely by using MEMS fabrication technology. Then, the channel surface was modified hydrophobically. Convex air bubbles naturally form in the hydrophobic cavities as soon as water is introduced.

To examine the effect of the convex air bubbles on the prevention of catheter blockage by microspheres, straight catheter channels were fabricated with and without micro-cavities. The length of the catheter channel was 40 mm. Because it is hard to specify the exact position where the blockage occurs, two pressure ports were connected to the catheter channel to detect the sudden pressure build-up by the blockage, and the distance between them was 32 mm. Microspheres with a diameter of 100 μm have been used clinically in embolization [25]. Thus, a channel height of 150 μm was designed by considering their size. Therefore, only one microsphere can be placed in the depth direction. Two-dimensional blockage arching can be situated. Additionally, the width of the catheter channel was varied and included 200, 300, and 400 μm, by considering the number of microspheres involved in the arching. Moreover, by considering the slip influence distance of a single bubble as 209 μm, which was reported in a previous work, two different intervals of 100 and 500 μm were used, indicating the distance between the cavities [23].

The catheter channel connected with the cavities and the pressure ports was fabricated by soft lithography. First, the mold pattern was prepared with photolithography. To make the mold pattern of the catheter channel, the adhesion promoter of HDMS was spin-coated on a 4-inch silicon wafer at 4000 rpm for 30 s. It was baked on a hot plate at 150 °C for 2 min. SU-8 100 photo-resist was coated at 500 rpm for 10 s and 1750 rpm for 30 s. It was cured on a hot plate at 65 °C for 10 min and at 95 °C for 30 min. Then, it was exposed to UV for 37 s for photolithography. Again, the exposed silicon wafer was placed on the hot plate and baked at 65 °C for 1 min and at 95 °C for 12 min [26]. The baked silicon wafer was cooled down and developed in a SU-8 developer until the catheter channel patterns were successfully obtained. Then, it was washed with isopropyl alcohol and de-ionized water several times and dried by heating it at 150 °C for 3 min.

Secondly, the PDMS(polydimethylsiloxane) microchannel catheter was fabricated by molding the PDMS. The inserted eyelets of the silicone tube were aligned and bonded to the inlet and outlet of the patterned SU-8 catheter channel and pressure ports, respectively. Then, the PDMS mixed with a pre-polymer and curing agent (Sylgard 184, Dow Corning, MI, USA) with a ratio of 10:1 was poured onto the patterned silicon wafer at a thickness of 5 mm. After removing the air bubbles remaining inside the PDMS in a vacuum oven, it was cured at 60 °C for 2 h to minimize the PDMS shrinkage.

Finally, each device was cut and exposed to oxygen plasma for 2 min with slide glass. This process makes the surface hydrophilic. Then, the mold PDMS and the slide glass were bonded naturally. The fabricated catheter channel was inspected with a microscope and SEM images. The inlet and outlet at both ends were successfully connected to the microchannel catheter as aligned. The height and width were measured to be 160 μm with a standard deviation of 3.68 μm and 426 μm with a standard of 3.71 μm, respectively. The cavity size was 106 μm with a standard deviation of 2.38 μm.

## 3. Clogging Prevention Experiment

In general, PVA (polyvinyl alcohol) or gelatin spheres are mostly used for the embolus microspheres. In this experiment, glass microspheres with a specific gravity of 2.5 were used due to the difficulty in the microscopic visualization of PVA spheres. The glass microspheres settled to the channel bottom and their movement was examined on the microscope. The size of the glass microspheres was from 90 to 106 μm and water were prepared as a carrier fluid. A high-speed camera (VEO 640L, Phantom, NJ, USA) was set up in a vertical microscope (IX 71, Olympus, Tokyo Japan) to visualize the movement of the microspheres and their blockage in the catheter microchannel. A syringe pump (PHD 2000, Harvard apparatus, Holliston, MA, USA) was connected to the inlet and introduced the microsphere-suspended water. The microspheres were injected randomly. It was hard to visualize the exact position of the occurrence of the blockage in the 40 mm-long channel in real-time due to the limitation of the microscope lens. Thus, to indicate the occurrence of the blockage, the differential pressure was measured between the two pressure ports. Moreover, the convex air bubbles were formed in the cavities by introducing pure de-ionized water in advance.

### 3.1. Slip Effect of the Convex Air Bubble

It was observed that the glass microspheres easily settled to the bottom of the catheter. Thus, in order to carry the microspheres stably in water, the Reynolds number of the flow was examined. By considering the hydraulic diameter of the catheter microchannel, Reynolds numbers ranging from 3.6 to 72.2 were tested, and it was found that microspheres could move stably with a Reynolds number over 20 [27].

The two straight microchannels with and without the air bubbles were compared. The channel width was 200 μm. The inlet flow rate was 257.1 μL/min, corresponding to an Re of 20. In the straight microchannel without the air bubbles, blockage occurred. It was hard to control the exact condition and position of the blockage occurrence. Only after the blockage occurred were its characteristics measured. Figure 2a shows a picture taken after the blockage occurred in the catheter channel without the air bubbles. Its angle was obtained from the line connecting the centers of the two microspheres and the channel length. The blockage arching angle was obtained at an angle of about 70° from the image analysis, similar to one theoretically obtained. However, it does not mean that the measured blockage angle of 70° is a threshold or minimal angle.

In the straight channel attached with the convex air bubbles formed in the microcavities, the microspheres slide freely without any blockage from the arching despite a geometrical arching angle of 85° due to the slippery and deformable surface of the convex air bubbles shown in Figure 2b.

Depending on the size of the catheter channel and the microspheres, the number of microspheres involved in the blockage arching can be determined. Figure 3 shows the 2-dimensional blockage arching involved with three microspheres in the straight catheter channel. The plane is positioned in the center where z equals 0 in height. The force diagram includes all the forces acting on the microsphere positioned in the middle and two microspheres pushed toward the wall. Three important forces are involved for the blockage arching. They are the drag force caused by the fluid flow, the friction force between the wall surface and the microsphere, and the reaction force between the two microspheres as a response to action and reaction.

The drag force was analyzed with the assumptions of a steady and incompressible flow. Additionally, a 2-dimensional fully developed velocity profile was considered along the catheter channel for the sake of convenience, even though the velocity profile is parabolic 3-dimensionally in the rectangular catheter channel with the sizes of the channel height of 160 μm, width of 426 μm, and length of 40 mm. In addition, the velocity profile could be affected by the existence of the microspheres. Due to a very low Reynolds number, a strong laminar flow is formed, and the microspheres move along the streamlines since the movement of the spheres is governed mainly by fluidic force. Therefore, in the calculation their effect on the velocity profile was also neglected.

The 2-dimensional velocity distribution was obtained on the plane where *y = 0* in the *z*-axis direction as Equation (1) [28,29]. The inlet flow rate, $Q_{in},$ was calculated by integrating the velocity distribution over the width as Equation (2).
(1)$$u_{z}\left(x,\;y=0\right)=\frac{16\alpha^{2}h^{2}}{\mu \pi^{4}}\left(\frac{\Delta p}{l}\right)\overset{\infty }{\underset{n=1}{\sum }}\overset{\infty }{\underset{m=1}{\sum }}\frac{\left(1-cos\;m\pi \right)\left(1-cos\;n\pi \right)}{mn\left(m^{2}+\alpha^{2}n^{2}\right)}sin\frac{n\pi }{2}sin\frac{m\pi }{2w}\left(x+w\right)$$
(2)$$Q_{in}=\int_{-w}^{w}u_{z}\left(x\right)dx=\frac{32\alpha^{3}h^{3}}{\mu \pi^{5}}\left(\frac{\Delta p}{l}\right)\overset{\infty }{\underset{n=1}{\sum }}\overset{\infty }{\underset{m=1}{\sum }}\frac{{\left(1-cos\;m\pi \right)}^{2}\left(1-cos\;n\pi \right)}{m^{2}n\left(m^{2}+\alpha^{2}n^{2}\right)}sin\frac{n\pi }{2}$$
where *α* is the aspect ratio of the catheter channel, μ is the viscosity of the fluid, and Δp is the pressure drop according to channel length, l. Then, the 2-dimensional velocity distribution can be expressed as Equation (3) by substituting Equation (2) into Equation (1).
(3)$$u_{z}\left(x\right)=Q_{in}\frac{\pi }{2w}\frac{\sum_{n=1}^{\infty }\sum_{m=1}^{\infty }\frac{\left(1-cos\;m\pi \right)\left(1-cos\;n\pi \right)}{mn\left(m^{2}+\alpha^{2}n^{2}\right)}sin\frac{n\pi }{2}sin\frac{m\pi }{2w}\left(x+w\right)}{\sum_{n=1}^{\infty }\sum_{m=1}^{\infty }\frac{{\left(1-cos\;m\pi \right)}^{2}\left(1-cos\;n\pi \right)}{m^{2}n\left(m^{2}+\alpha^{2}n^{2}\right)}sin\frac{n\pi }{2}}$$

Assume that two microspheres attached on the wall surface are placed symmetrically, and the flow rates on them, $Q_{l}$ and $Q_{u},$ are obtained by subtracting the flow rate on the middle microsphere, $Q_{c}$, from the inlet flow rate as Equation (4).
(4)$$\begin{array}{c} Q_{l}=Q_{u}=\frac{Q_{in}-Q_{c}}{2}\\ =\frac{Q_{in}}{2}\left[1-\frac{\sum_{n=1}^{\infty }\sum_{m=1}^{\infty }\frac{\left(1-cos\;m\pi \right)\left(1-cos\;n\pi \right)}{m^{2}n\left(m^{2}+\alpha^{2}n^{2}\right)}sin\frac{n\pi }{2}\left[cos\frac{m\pi }{2w}\left(w-\frac{a}{2}\right)-cos\frac{m\pi }{2w}\left(w+\frac{a}{2}\right)\right]}{\sum_{n=1}^{\infty }\sum_{m=1}^{\infty }\frac{{\left(1-cos\;m\pi \right)}^{2}\left(1-cos\;n\pi \right)}{m^{2}n\left(m^{2}+\alpha^{2}n^{2}\right)}sin\frac{n\pi }{2}}\right] \end{array}$$

The average velocity is then obtained by dividing the flow rate entering into each microsphere with the confronted projection area. The average velocity on the center microsphere, ${\bar{V}}_{c}$, is calculated in Equation (5).
(5)$$\begin{array}{c} {\bar{V}}_{c}=\frac{Q_{c}}{a}\\ =\frac{\sum_{n=1}^{\infty }\sum_{m=1}^{\infty }\frac{\left(1-cos\;m\pi \right)\left(1-cos\;n\pi \right)}{m^{2}n\left(m^{2}+\alpha^{2}n^{2}\right)}sin\frac{n\pi }{2}\left[cos\frac{m\pi }{2w}\left(w-\frac{a}{2}\right)-cos\frac{m\pi }{2w}\left(w+\frac{a}{2}\right)\right]}{a\sum_{n=1}^{\infty }\sum_{m=1}^{\infty }\frac{{\left(1-cos\;m\pi \right)}^{2}\left(1-cos\;n\pi \right)}{m^{2}n\left(m^{2}+\alpha^{2}n^{2}\right)}sin\frac{n\pi }{2}} \end{array}$$

Because the open area to two microspheres is (2w−a), the position of their average velocity does not coincide with the geometric center of the microspheres. The position of the average velocity of the microsphere on the wall is obtained by dividing the flow rate with the open area, as shown in Equation (6).
(6)$${\bar{V}}_{l}={\bar{V}}_{u}=\frac{Q_{l}}{\frac{2w-a}{2}}=\frac{Q_{in}}{2w-a}\left[1-\frac{\sum_{n=1}^{\infty }\sum_{m=1}^{\infty }\frac{\left(1-cos\;m\pi \right)\left(1-cos\;n\pi \right)}{m^{2}n\left(m^{2}+\alpha^{2}n^{2}\right)}sin\frac{n\pi }{2}\left[cos\frac{m\pi }{2w}\left(w-\frac{a}{2}\right)-cos\frac{m\pi }{2w}\left(w+\frac{a}{2}\right)\right]}{\sum_{n=1}^{\infty }\sum_{m=1}^{\infty }\frac{{\left(1-cos\;m\pi \right)}^{2}\left(1-cos\;n\pi \right)}{m^{2}n\left(m^{2}+\alpha^{2}n^{2}\right)}sin\frac{n\pi }{2}}\right]$$

The drag force exerted on the microsphere is known as the Stokes force in a low Reynolds flow. So, each drag force of $D_{c}$, $D_{l},$ and $D_{u}$ is given by Equations (7) and (8), where *μ* is the dynamic viscosity of the fluid.
(7)$$D_{c}=6\pi \mu \left(\frac{a}{2}\right){\bar{V}}_{c}$$
(8)$$D_{l}=D_{u}=6\pi \mu \left(\frac{a}{2}\right){\bar{V}}_{l}=6\pi \mu \left(\frac{a}{2}\right){\bar{V}}_{u}$$

The center of the drag force is assumed to be identical to the center of the mean velocity. By considering the drag force and the center position, the blockage arching angle, θ, can be obtained from Equation (9).
(9)$$\theta =sin^{-1}\frac{2w-a}{2a}$$

The action-reaction force of $F_{r}$ can be obtained from the drag force, $D_{c}$, acting on the microsphere positioned in the middle and the blockage arching angle, θ.
(10)$$F_{r}=\frac{D_{c}}{2cos\theta }$$

Finally, the friction force of $F_{f}$ is obtained as follows.
(11)$$F_{r}=\mu_{f}F_{N}=\mu_{f}F_{r}sin\theta$$
where $\mu_{f}$ is the coefficient of the static friction. In general, the static friction is larger than the dynamic friction. The three important forces to form the blockage arching of microspheres in the catheter channel were calculated. For $\mu_{f}$ = 0.7, a = 100–145 μm, w = 150 μm, α = 2.5, and μ = 1.12 × 10^−3^ N/m∙s, the blockage arching angle was analyzed. If the summation of drag forces is smaller than the friction force, it is possible for a blockage to occur. Figure 4 shows the blockage arching angle, and it was determined to be over 76°, approximately. In addition, the blockage angle depends on the material of the microspheres because the coefficient of the static friction differs by material.

In the visualization, the microsphere blockage occurred at an arching angle of 70° in the straight catheter channel without the convex air bubbles, while the microspheres moved freely in the straight catheter channel with the convex air bubbles, despite a geometrical arching angle of over 85° being formed. To explain the slip effect of the convex air bubble on the prevention of the microsphere blockage, the possible range of the blockage arching angle was calculated for the different friction coefficients. Figure 5 shows that the range of the blockage arching angle also decreases as the friction coefficient decreases. As a result, it is concluded that the slip effect on the convex air bubble can reduce the friction and can prevent the blockage by the arching of the microspheres in the catheter channel.

### 3.2. Centrifugal Effect on the Convex Air Bubble

The centrifugal effect of the convex air bubble on the drift distance was also examined. The channel width was 300 μm, and the interval of the convex air bubbles was 500 μm. The experiment was performed at a flow rate of 300 μL/min (Re = 20) in the catheter microchannel with a width of 300 μm. As the microspheres passed through four consecutive convex air bubbles placed within a channel length of 2400 μm, the trajectories of the microspheres were analyzed [Appendix A].

Figure 6 visualizes two trajectories of the microspheres moving around the upper and lower convex air bubbles attached in the straight microchannel. They were obtained by averaging the movements of five different microspheres. The microspheres gradually moved to the middle of the channel by passing through the convex air bubbles. Near the fourth convex air bubble in the flow direction, the drift distances of the upper and lower microspheres from their initial positions were measured to be 238 and 166.8 μm, respectively. Additionally, their drift distances away from their initial positions were analyzed for two different sizes of convex air bubbles. The channel width was 400 μm, and the interval of the convex air bubbles was 500 μm. Figure 7 plots the positions of the microspheres passing around the upper and lower convex air bubbles with a height of 60 (blue asterisk) and 102 μm (red diamond), respectively. When the microspheres passed the convex air bubbles with a height of 60 μm, they moved 64.6, 77.7, 102.9, and 134.7 μm upward gradually from their initial positions and converged to the center of the catheter channel. Moreover, for a bubble height of 102 μm, they drifted 103.9, 105.6, 102.2, and 108.8 μm, sequentially. The microspheres passing around the convex air bubbles drifted rapidly due to a large centrifugal acceleration that is inversely linear to the bubbles small radius of curvature. After they were placed away from the convex air bubbles, their drift distance was decreased because the centrifugal effect became weakened.

In addition, the effect of the interval of the convex air bubbles on the drift distance was examined. Convex air bubbles with two different intervals of 500 and 100 μm were compared. The heights of the air bubbles were measured to be 59.3 and 45.4 μm, respectively. The positions of the microspheres were measured along the catheter channel by choosing the microspheres that were initially placed near the wall surface. Figure 8 shows that the microspheres drifted upward 90.6, 96.8, 106.2, and 115.7 μm passing gradually through each convex air bubble for an interval of 100 μm, while they moved upward 64.6, 77.7, 102.9, and 134.7 μm for an interval of 500 μm. On the convex air bubbles with a short interval, they rose more quickly and were focused into the center. Figure 9 shows the attack angle of the microspheres when they moved to the second convex air bubble just after passing through the first convex air bubble. The attack angle was about 3° for the small interval of 100 μm and about 29° for the interval of 500 μm.

When the microspheres passed around the first convex air bubble, the large air bubble produced more drift distance so that the microspheres moved to the center of the catheter channel. Additionally, the short interval of the convex air bubbles prevented the microspheres from approaching the wall, which might cause the blockage arching of the microspheres in the channel. Thus, it is advantageous to design a straight catheter channel with large cavities to form the large convex air bubbles and with a small interval to prevent blockage by arching.

The drift distance obtained by the centrifugal effect can be theoretically explained. Figure 10 shows the movement of the microsphere on the convex air bubble. The radius of the convex air bubble is very small so that a large centrifugal acceleration can be generated, and a large drift distance on the surface of the convex air bubble in the normal direction can be produced.

The centrifugal force exerted on the microsphere is given by Equation (12), where m is the mass of the microsphere, U is the mean velocity of the flow, and r is the radius of curvature of the convex air bubble.
(12)$$F_{c}=\frac{mU^{2}}{r}=\frac{\rho \pi a^{3}}{6}\frac{mU^{2}}{r}$$

This centrifugal force enables the microsphere to escape from the surface of the air bubble in its normal direction. The drag force resists against the movement of the microsphere in the normal direction. It can be obtained by the Stokes force where *V*_*s*_ is the drift velocity in the normal direction.
*F*_*D*_ = 3*π**μ**a**V*_*s*_(13)

In a low Reynolds flow, the viscous effect is dominant compared with the inertia effect. Thus, it reaches a terminal velocity very soon, and the two forces are balanced. Then, the drift distance can be calculated by the drift velocity and travel time. The drift velocity was calculated from the balance between the drag force and the centrifugal force as follows.
(14)$$V_{s}=\frac{\rho a^{2}U^{2}}{18\mu r}$$

Then, the traveling time t is equal to the time that the fluid takes to travel around the convex air bubble with a velocity of *U*. The traveling length is assumed to be a half circle.
(15)$$t=\frac{\pi r}{U}$$

Finally, the drift distance can be obtained by multiplying the drift velocity and travel time.
(16)$$s=V_{s}t=\frac{\pi \rho a^{2}U}{18\mu }$$

The centrifugal effect of the convex air bubble could drift the microspheres away from the wall surface when passing over the convex air bubble. As a result, the drift distance can prevent the formation of blockage by arching.

Aside from slip and centrifugal effect on the prevention of the blockage, the elastic deformation of the air bubble can be considered additionally. The air bubble can be deformed by the microsphere pushed by the fluidic force in a low Reynolds number flow, what is called Stokes force and a surface force. The air bubble can resist to the deformation of the convex meniscus of the air bubble formed due to surface tension, which is a line force. The ratio of the fluidic force and capillary pressure force becomes zero when the deformation size of the air bubble of a characteristic size is very small. It means that the fluidic force is negligible compared with the capillary pressure force. This scale effect can explain why the elasticity effect of the air bubble on the blockage is negligible ([30], Appendix A).

## 4. Conclusions

The effect of the convex air bubbles on the prevention of blockages by embolic microspheres in catheters was experimentally evaluated and theoretically explained. The catheter channel was fabricated with PDMS using the MEMS fabrication process. It was shaped with microcavities to form convex air bubbles and two pressure ports to signal the occurrence of a blockage. The width, cavity, and interval between the cavities were varied to examine the effect of the prevention of microsphere blockage. In the straight catheter channel without air bubbles, blockage arching occurred at an angle of about 76°, whereas in the channel with convex air bubbles formed in the cavities, the microspheres slid freely without any blockages, even at a geometrical arching angle of 85°, due to the slippery and deformable surface of the convex air bubbles. It was also theoretically explained that the reduced friction by the slip could increase the blockage arching angle.

In addition, the convex air bubble produced large centrifugal acceleration obtained from its small radius of curvature, and this acceleration was exerted on the microspheres while they were traveling around the convex air bubbles. As a result, the centrifugal acceleration made the microspheres drift away from the wall of the catheter channel. Thus, this reduced their chance to attach onto the wall, resulting in the prevention of microsphere blockage by arching.

From the experimental results, large convex air bubbles and a small interval between the cavities were effective at preventing microsphere blockage. Thus, it can be concluded that it will be advantageous to design a catheter channel with large cavities and a small interval for the prevention of microsphere blockage in embolization. For further study, the occurrence of blockages from the bending of the catheter, which often happens during embolization, will be examined by considering the radius of the curvature of the catheter.

## Figures and Tables

**Figure 1 micromachines-11-01040-f001:**
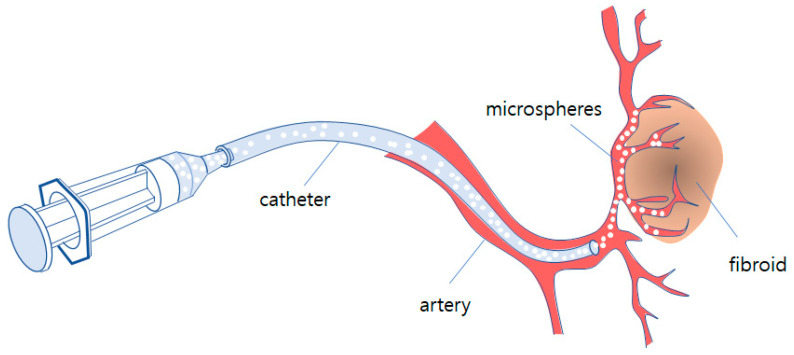
Schematic of fibroid embolization of microspheres with catheter.

**Figure 2 micromachines-11-01040-f002:**
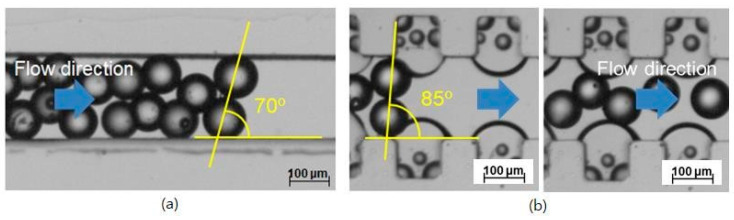
Prevention of microsphere blockage in the catheter channel by convex air bubbles; (**a**) Occurrence of the blockage arching at an arching angle of 70° in the straight catheter channel without the convex air bubbles, (**b**) Free movement of the microspheres at a geometrical arching angle of over 85° in the straight catheter channel with the convex air bubbles.

**Figure 3 micromachines-11-01040-f003:**
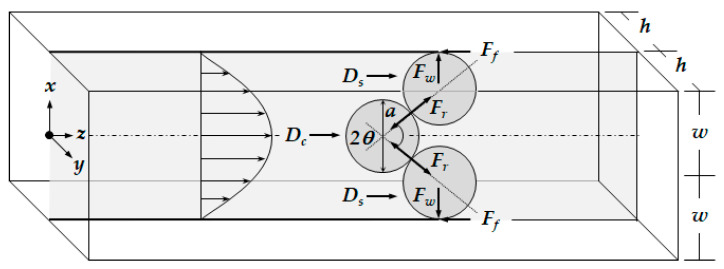
Diagram of the forces acting on the microspheres when the blockage occurs in catheter by arching.

**Figure 4 micromachines-11-01040-f004:**
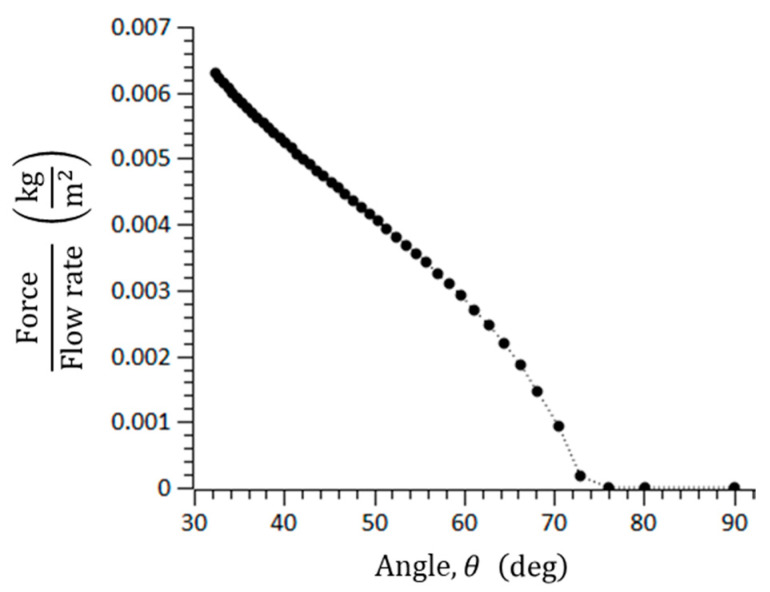
Threshold angle to cause geometrical arching in a rectangular microchannel.

**Figure 5 micromachines-11-01040-f005:**
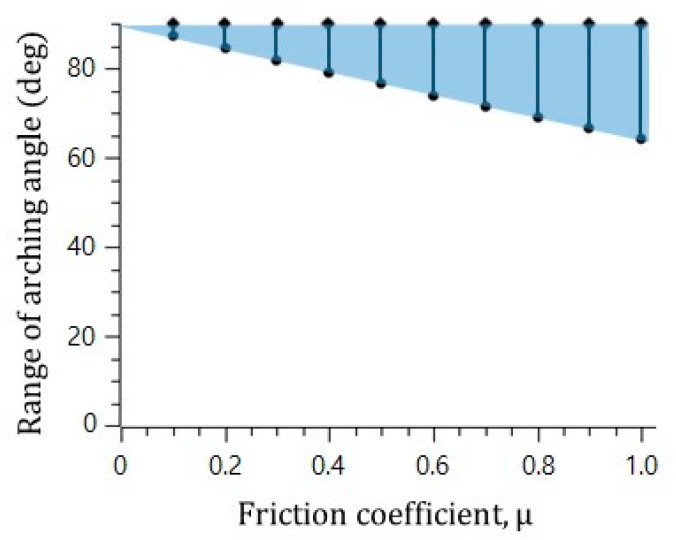
Decrease in the range of arching angle with decreasing friction force.

**Figure 6 micromachines-11-01040-f006:**
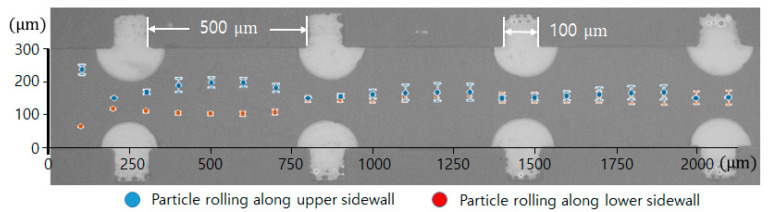
Trajectory of microsphere movement over the air bubbles in the catheter channel.

**Figure 7 micromachines-11-01040-f007:**
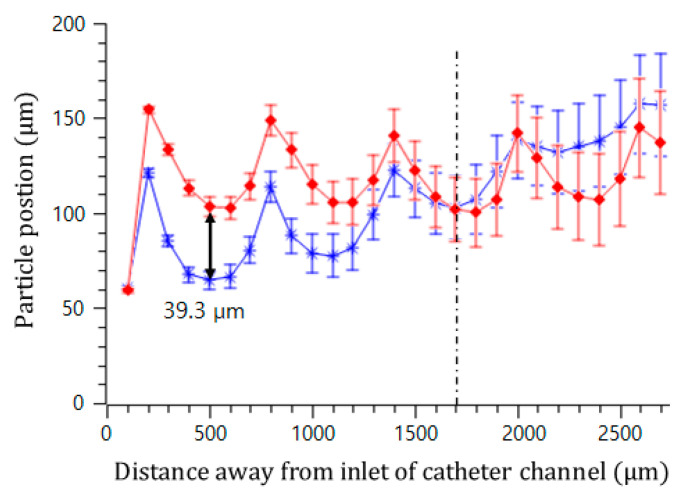
Trajectory of microsphere movement in the catheter channel with a bubble height of 102 μm (red diamond) and a bubble height of 60 μm (blue asterisk).

**Figure 8 micromachines-11-01040-f008:**
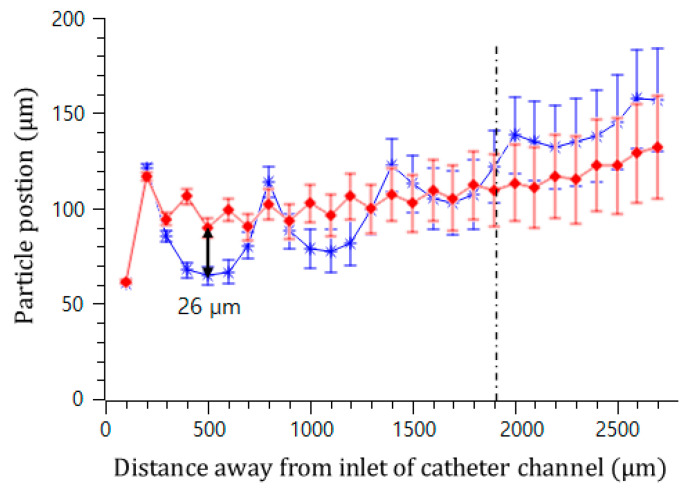
Trajectory of microsphere movement in the bubble catheter channel with the interval lengths of 100 μm (red diamond) and 500 μm (blue asterisk).

**Figure 9 micromachines-11-01040-f009:**
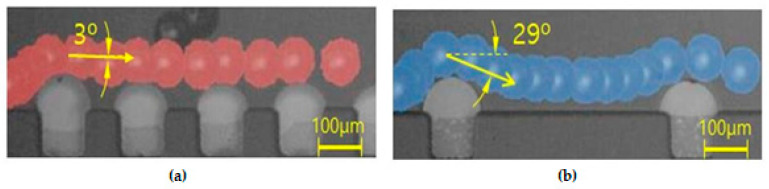
Attack angle on the successive air bubble for the different intervals: (**a**) the interval of the successive air bubble of 100 μm and (**b**) the interval of the successive air bubble of 500 μm.

**Figure 10 micromachines-11-01040-f010:**
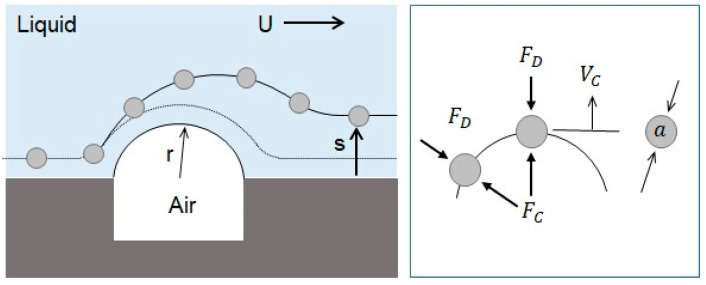
Drift of microspheres away from wall caused by centrifugal force.

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
