# Peer review of "Prevention of Microsphere Blockage in Catheter Tubes Using Convex Air Bubbles"

_micromachines, 2020, doi:10.3390/mi11121040_

Round 1

Reviewer 1 Report

The authors describe the design of catheter tubes using convex air bubbles to prevent blockage of microspheres. The manuscript details the appropriate scientific methods and results. It can benefit from the following. 

  1. On page 3 line 119, it would be appropriate to indicate the standard deviation of the height and width of the fabricated channel as opposed to the error rate. The values are the variation of the fabrication process rather than an error. 
  2. Does the material of the microsphere have an effect on blockage? It would be useful to compare different materials or provide justification for applications. 
  3. Discussion on the practical limitations to create cylindrical tube catheters is useful including the challenges to create air bubbles. As this study demonstrated on a 2D channel. 
  4. Physical storage of the channel after buble formation needs to be discussed. once the bubbles generated, will be functional indefinitely? or the bubbles need to be generated every time?

Author Response

Dear editor,

The authors really appreciate to the reviewers’ effort on managing our paper to improve the quality of the manuscript. Also, we have tried to do our best to address all the comments in the revised manuscript.

Manuscript ID: micromachines-987396

Title: Prevention of microsphere blockage in catheter tubes using convex air bubbles

Authors: Yeun Jung Jung, Jeung Sang Go *, Dong Hyeok Park, Sandoz John Kinson Steve Jeo Kins, Young Deok Kim

Following pages are the responses to the reviewers’ comments.

Sincerely yours,

Jeung Sang Go, Ph.D.

Prof. School of Mechanical Engineering, Pusan National University

2, 63beongil, Geumjeong-gu, Busan 46241, Republic of Korea

Phn) +82-51-510-3512/Email: [email protected]/Web: nextmems.pusan.ac.kr

Response to the reviewer 1.

The authors describe the design of catheter tubes using convex air bubbles to prevent blockage of microspheres. The manuscript details the appropriate scientific methods and results. It can benefit from the following.

Comment 1. On page 3 line 119, it would be appropriate to indicate the standard deviation of the height and width of the fabricated channel as opposed to the error rate. The values are the variation of the fabrication process rather than an error.

(Answer) Appreciation. The errors are corrected with STD. Please refer line 127~129, page 3.

Comment 2. Does the material of the microsphere have an effect on blockage? It would be useful to compare different materials or provide justification for applications.

(Answer) The authors totally agree with the reviewer’s comment. The material of microspheres effects on the blockage because the static friction coefficient is different from materials as expressed in Eq. 11. In the experiment, glass spheres with a specific gravity of 2.4 were used and they were sedimented on the bottom surface in the channel. As a result, the 2D blockage could be experimented, similar to the theoretical situation. The authors apologize that this is referred in the revised manuscript instead of repeating the experiment. Please refer line 131~132, page 3 and line 243, page 7, respectively.

Comment 3. Discussion on the practical limitations to create cylindrical tube catheters is useful including the challenges to create air bubbles. As this study demonstrated on a 2D channel.

(Answer) The authors agree with the reviewer’s comment because the catheters for embolization are generally cylindrical. Also, making the cylindrical tube with cavities on the inner surface of the tube will be very challenging.

The hydrophobic cylindrical tubes are manufactured by extrusion process. However, in order to form the air bubble generation for the introduction of a liquid the tube must have cavities on the inner surface of the tube. This makes the manufacturing process very complicated. This content is included in the revised manuscript. Please refer line 87~93, page 2.

Comment 4. Physical storage of the channel after bubble formation needs to be discussed. once the bubbles generated, will be functional indefinitely? or the bubbles need to be generated every time?.

(Answer) The authors apologize that the reviewer may misunderstand the storage of the channel after bubble generation. In general, the catheter for medical embolization is disposable. The bubbles in the cavities are formed naturally as soon as a liquid suspended with microspheres is introduced into the catheter tube. So, it is not necessary that the tube formed with air bubbles is stored. Also, the tube can be stored in a N2–filled vinyl package before practical use.

Reviewer 2 Report

The paper reports convex air bubbles attached to the catheter tubes' surface to prevent the catheter channel's blockage. Meanwhile, the authors provide the potential mechanism to explain the phenomenon via the theoretical studies on the geometrical arching of the microspheres, the slip effect, and the centrifugal effect of the air bubbles. Some parts of the work are nicely done and scientifically sound. However, some major modifications need to be made before the publishing of the paper. My detailed comments are below:

  1. Do the sizes of the microspheres and the catheter influence the results? It seems that the authors adjusted the relationship between the depth of the catheter and the diameter of microspheres to get the phenomenon they want. This is essential because the diameter of the catheter is unchangeable, which depends on the clinical circumstance. And the most crucial problem is that the catheter is long-strip-shaped, which conflicts with the real cases.
  2. Do the authors take into account the properties of the clogging particles? Such as the hardness, shape, size, concentration of the clogging particles. Because the clogging particles' properties are unpredictable, the authors should add more parameters of the clogging particles to simulate embolization.
  3. The expression claimed that "only two mechanisms behind the clogging in the catheter at Line 43" are not accurate? How about biological mechanisms, like infection, thrombogenesis? And fluid dynamics influences?
  4. Many expressions in the paper are not understandable. They are very confusing to the readers.

 For example,

  1. Line 92, the width of WHAT?
  2. Line 269, the size of WHAT?

  1. The schematic of the catheter is recommended to be presented in the paper.

  1. It is necessary to check the format, grammar, typos in the paper.

   For example,

  1. Line 109, "tubeinserted" should add a space in between.
  2. Line 111~112 and Line 257-258, the line feeds are not suitable.

  1. The reference of "Microspheres with a diameter 89 of 100 μm have been used clinically in embolization" at Line 89 should be added.

Author Response

Dear editor,

The authors really appreciate to the reviewers’ effort on managing our paper to improve the quality of the manuscript. Also, we have tried to do our best to address all the comments in the revised manuscript.

Manuscript ID: micromachines-987396

Title: Prevention of microsphere blockage in catheter tubes using convex air bubbles

Authors: Yeun Jung Jung, Jeung Sang Go *, Dong Hyeok Park, Sandoz John Kinson Steve Jeo Kins, Young Deok Kim

Following pages are the responses to the reviewers’ comments.

Sincerely yours,

Jeung Sang Go, Ph.D.

Prof. School of Mechanical Engineering, Pusan National University

2, 63beongil, Geumjeong-gu, Busan 46241, Republic of Korea

Phn) +82-51-510-3512/Email: [email protected]/Web: nextmems.pusan.ac.kr

Response to the reviewer 3.

The paper reports convex air bubbles attached to the catheter tubes' surface to prevent the catheter channel's blockage. Meanwhile, the authors provide the potential mechanism to explain the phenomenon via the theoretical studies on the geometrical arching of the microspheres, the slip effect, and the centrifugal effect of the air bubbles. Some parts of the work are nicely done and scientifically sound. However, some major modifications need to be made before the publishing of the paper. My detailed comments are below:

Comment 1. Do the sizes of the microspheres and the catheter influence the results? It seems that the authors adjusted the relationship between the depth of the catheter and the diameter of microspheres to get the phenomenon they want. This is essential because the diameter of the catheter is unchangeable, which depends on the clinical circumstance. And the most crucial problem is that the catheter is long-strip-shaped, which conflicts with the real cases.

(Answer) The size of microspheres ranging from 90 to 106 μm is very common size for embolization. The most problematic size of the catheter is a long and thin catheter with a diameter of less than 1 mm. The authors write this in the revised manuscript. Please refer Figure 1 and line 38, page 1.

Comment 2. Do the authors take into account the properties of the clogging particles? Such as the hardness, shape, size, concentration of the clogging particles. Because the clogging particles' properties are unpredictable, the authors should add more parameters of the clogging particles to simulate embolization.

(Answer) The authors agree with the reviewer’s comment on considering other properties of the clogging particles. In general, the microspheres should be formed perfect sphere for their flowability in artery. Otherwise, it is hard to be applied for embolization. Also, two major materials of PVA and gelatine microspheres are commercialized. Their hardness should be controlled for purpose. In the experiment, we used glass microspheres and took into account spherical shape, a very hard stiffness, and static friction coefficient as a surface property. Please refer line 131~132, page 3.

Comment 3. The expression claimed that "only two mechanisms behind the clogging in the catheter at Line 43" are not accurate? How about biological mechanisms, like infection, thrombogenesis? And fluid dynamics influences?

(Answer) The authors apologize inaccurate expression to make the reviewer to misunderstand. The clogging of microspheres is occurred not in artery but in the catheter tube. So, the clogging mechanism is explained mainly by mechanical arching and electrical deposition rather than by biological involvement. Please refer line 48~50, page 2.

Comment 4. Many expressions in the paper are not understandable. They are very confusing to the readers.

For example,

1.Line 92, the width of WHAT?

2.Line 269, the size of WHAT?

(Answer) The authors apologize confusing expressions. The expressions are carefully checked again for potential readers. Many thanks. Please refer line 102~103, page 3 and 290~291, page 9.

Comment 5. The schematic of the catheter is recommended to be presented in the paper.

(Answer) Appreciation to this comment. The schematic drawing of the catheter for fibroid embolization is included newly in the revised manuscript. Please refer Figure 1.

Comment 6. It is necessary to check the format, grammar, typos in the paper.

   For example,

1.Line 109, "tubeinserted" should add a space in between.

2.Line 111~112 and Line 257-258, the line feeds are not suitable.

(Answer) Thank you very much. The authors also find typos and formats. They may be conversion mistake. The format, grammar, typos in the paper are checked carefully and corrected.

Comment 7. The reference of "Microspheres with a diameter 89 of 100 μm have been used clinically in embolization" at Line 89 should be added.

(Answer) The authors thank you for this. The reference is included. Please refer in line 100, page 3.

Reviewer 3 Report

The manuscript presented by Jung and co-authors introduced a simple yet ingenious design to prevent microsphere blockage in catheter tubes based on trapped bubbles in side cavities. Overall, the manuscript is easy to follow, the theory behind the phenomenon is quite confusing. Please see below for my concerns:

1) The authors mentioned the slip effect of bubble surface, yet bubbles are elastic. Please explain why elasticity is not considered and why the centrifugal force is the reason for microsphere to pass through bubbles. If the elasticity or bouncing forces are not the reasons, is it possible replace bubbles with semispherical geometry though non-slip conditions should be considered.

2) In line 231, please explain how to obtain a degree of 70 °, is it the minimal angle for blockage? Was this the only angle the authors obtained, or the angle was obtained from a series of experiments with a threshold of 70 °.

3) In line 181, explain why area is a. In line 185, explain why the center position of the average velocity is zero.

4) In line 164, I believe the flow is not the Couette flow, which involves a moving plate and a fixed plate. The flow velocity profile should be parabolic in width and height direction in a rectangular cross section channel, but the author did not explain how they obtain the equation 1.

5) Please explain why the microspheres did not affect the flow velocity profile.

Author Response

Dear editor,

The authors really appreciate to the reviewers’ effort on managing our paper to improve the quality of the manuscript. Also, we have tried to do our best to address all the comments in the revised manuscript.

Manuscript ID: micromachines-987396

Title: Prevention of microsphere blockage in catheter tubes using convex air bubbles

Authors: Yeun Jung Jung, Jeung Sang Go *, Dong Hyeok Park, Sandoz John Kinson Steve Jeo Kins, Young Deok Kim

Following pages are the responses to the reviewers’ comments.

Sincerely yours,

Jeung Sang Go, Ph.D.

Prof. School of Mechanical Engineering, Pusan National University

2, 63beongil, Geumjeong-gu, Busan 46241, Republic of Korea

Phn) +82-51-510-3512/Email: [email protected]/Web: nextmems.pusan.ac.kr

Response to the reviewer 2.

The manuscript presented by Jung and co-authors introduced a simple yet ingenious design to prevent microsphere blockage in catheter tubes based on trapped bubbles in side cavities. Overall, the manuscript is easy to follow, the theory behind the phenomenon is quite confusing. Please see below for my concerns:

Comment 1. The authors mentioned the slip effect of bubble surface, yet bubbles are elastic. Please explain why elasticity is not considered and why the centrifugal force is the reason for microsphere to pass through bubbles. If the elasticity or bouncing forces are not the reasons, is it possible replace bubbles with semi-spherical geometry though non-slip conditions should be considered.

(Answer) Scale effect can be explained by considering the change in the ratio of two different effects as a characteristic size decreases. The force to deform the bubble can be obtained by fluidic force, what is called Stokes force and acts on its surface, which is a surface effect. Whereas, the bubble resists to this deformation force by capillary pressure force, which is a line force. To deform the bubble, a small radius of the deformation curvature should occur in advance, which is a very large capillary pressure force. It means that it is very hard to deform the bubble with the spheres pushed by the fluidic force. As a result, the bubbles slip to move before the elastic deformation occurs. Please refer line 353 ~ 360, page 11.

Comment 2. In line 231, please explain how to obtain a degree of 70°, is it the minimal angle for blockage? Was this the only angle the authors obtained, or the angle was obtained from a series of experiments with a threshold of 70 °.

(Answer) The authors apologize confusion. As a matter of fact, it is hard to control condition and position of the blockage occurrence. Only after the blockage occurs, the blockage characteristic can be examined. Figure 1(a) shows the picture taken after the blockage occurred in the catheter channel. Its angle was obtained from the line connecting the centers of the two spheres and the channel length and the theoretical blockage angle was obtained at over about 70° as shown in Figure 3. Because many mechanisms are involved when the blockage of the spheres occurs, the measured blockage angle of 70° can not be a threshold or minimal angle. So, that’s why the authors write the measured angle only. Pease refer line 154~157, page 4.

Comment 3. In line 181, explain why area is a. In line 185, explain why the center position of the average velocity is zero.

(Answer) Apology for confusion. The authors think that this comment is also related with the comment 4 and 5. As the reviewer comments in the comment 5 as well, they may affect the velocity profile. We apologize that the velocity profile around the moving sphere in the channel has not measured. But due to a very low Reynolds number flow, it is formed a strong laminar flow and the microspheres move along the streamlines since the movement of the spheres is governed mainly by fluidic force. The coordinate of (x,y) is placed in the center of the channel and thus the center position of the average is assumed to be zero. The authors refer this in the revised manuscript. Please refer line 177, page 5 and 207~208, page 6.

Comment 4. In line 164, I believe the flow is not the Couette flow, which involves a moving plate and a fixed plate. The flow velocity profile should be parabolic in width and height direction in a rectangular cross section channel, but the author did not explain how they obtain the equation 1.

(Answer) The authors totally agree with the reviewer’s comment. The Couette flow can be obtained for the constant cross-sectional channel formed by two long, straight and parallel plates. The sizes of the fabricated catheter channel are 160 μm-high, 426 μm-wide and 40 mm-long with a constant cross-section. So, the velocity profile should be parabolic in width and height. The velocity of “u” in the (Eq. 1) is brought from the Couette flow to calculate the blockage angle roughly only for convenience. The authors refer this limitation in the revised manuscript. Please refer line 180~188, page 5.

Comment 5. Please explain why the microspheres did not affect the flow velocity profile.

(Answer) The authors apologize that the effect of the microspheres on the flow velocity profile has not been measured in this work. It will be very challenging. We also think that the flow velocity profile may be affected by the existence of the microspheres. As answered to the comment 3, due to a very low Reynolds number flow, a strong laminar flow is formed and the microspheres move along the streamlines since the movement of the spheres is governed mainly by fluidic force. Please refer line 180~188, page 5.

Round 2

Reviewer 2 Report

The authors have edited the manuscript and make it easier for the readers to follow. I suggest the paper to be published.

Reviewer 3 Report

Thanks for the authors’ responses. The manuscript has been improved, yet please answer the following questions:

  • The authors mentioned that “Scale effect can be explained by considering the change in the ratio of two different effects as a characteristic size decreases. The force to deform the bubble can be obtained by fluidic force, what is called Stokes force and acts on its surface, which is a surface effect.”. However, I did see bubble deformation in Figure 2b. I understand that the fluid force is negligible, yet the microspheres were directly in contact with the bubbles. Please explain why bubble deformation is not considered. Also, please cite related references.
  • Line 186-188. I am not sure why the authors still use Couette flow in the manuscript, why the profile in a rectangular cross section channel is not used? Also, the velocity profile in the Couette flow is linear. If possible, please cite the reference for the given equation.
  • It would be better to add a video clip demonstrating the effect of bubbles.

Round 3

Reviewer 3 Report

The manuscript has been significantly improved. Thank you. I recommend the publication as it is.